# Laboratory Investigation of Carbon Black/Bio-Oil Composite Modified Asphalt

**DOI:** 10.3390/ma14174910

**Published:** 2021-08-29

**Authors:** Ping Zhang, Lan Ouyang, Lvzhen Yang, Yi Yang, Guofeng Lu, Tuo Huang

**Affiliations:** 1Test and Detection Centre for Highway Engineering, Changsha University of Science & Technology, Changsha 410114, China; zhangping@csust.edu.cn (P.Z.); yangyi@csust.edu.cn (Y.Y.); gonganluguofeng@163.com (G.L.); huangtuomao@163.com (T.H.); 2Hunan Communication Engineering Polytechnic, Changsha 410132, China; sup721@163.com; 3Xiandai Investment Co., Ltd., Changsha 410029, China

**Keywords:** RSM, road engineering, carbon black, bio-oil, composite modified asphalt, asphalt mixture

## Abstract

As environmentally friendly materials, carbon black and bio-oil can be used as modifiers to effectively enhance the poor high-temperature and low-temperature performance of base asphalt and its mixture. Different carbon black and bio-oil contents and shear time were selected as the test influencing factors in this work. Based on the Box–Behnken design (BBD), carbon black/bio-oil composite modified asphalt was prepared to perform the softening point, penetration, multiple stress creep and recovery (MSCR), and bending beam rheometer (BBR) tests. The response surface method (RSM) was used to analyze the test results. In addition, the base asphalt mixtures and the optimal performance carbon black/bio-oil composite modified asphalt mixtures were formed for rutting and low-temperature splitting tests. The results show that incorporating carbon black can enhance the asphalt’s high-temperature performance by the test results of irrecoverable creep compliance (*J_nr_*) and strain recovery rate (*R*). By contrast, the stiffness modulus (*S*) and creep rate (*M*) test results show that bio-oil can enhance the asphalt’s low-temperature performance. The quadratic function models between the performance indicators of carbon black/bio-oil composite modified asphalt and the test influencing factors were established based on the RSM. The optimal performance modified asphalt mixture’s carbon black and bio-oil content was 15.05% and 9.631%, and the shear time was 62.667 min. It was revealed that the high-temperature stability and low-temperature crack resistance of the carbon black/bio-oil composite modified asphalt mixture were better than that of the base asphalt mixture because of its higher dynamic stability (*DS*) and toughness. Therefore, carbon black/bio-oil composite modified asphalt mixture can be used as a new type of choice for road construction materials, which is in line with green development.

## 1. Introduction

Asphalt pavement is diffusely used in the infrastructure construction of countries all over the world. However, with the increase in vehicle load and traffic volume, the asphalt pavement has shown more problems, such as rutting [1], fatigue cracking [2], and water damage [3]. Therefore, it cannot satisfy the current people’s living standards. From the point of view of material composition, asphalt binder is an essential part of the asphalt mixture, and its performance directly affects the service level of asphalt pavement. Many studies [4,5,6,7] have shown that the asphalt’s rheological properties and low-temperature properties can be improved by incorporating polymers and nanomaterials into the base asphalt. Then, the asphalt mixture’s low-temperature crack resistance and high-temperature stability were also improved. However, these modifiers often have high production costs and complex production processes, which do not conform to the concept of green economic development [8].

In recent years, incorporating inorganic minerals as modifiers into base asphalt has received extensive attention from researchers. These materials are often low in price, simple in production technology, and have a wide range of sources, mainly including carbon black [9], cement [10], lime [11], and diatomite [12]. As a carbon-derived material, carbon black can be used as a modifier to change the asphalt’s internal structure, thereby improving the asphalt’s high-temperature performance and resistance to plastic deformation and elastic properties [13]. Cong et al. [14] added three different types of carbon black to SBS modified asphalt. Through experiments, it was obtained that the addition of three carbon blacks can improve the SBS modified asphalt’s anti-aging performance and high-temperature performance, while it will reduce the asphalt’s storage stability. Tacettin et al. [15] studied the effects of carbon black modifiers on the asphalt’s high-temperature and low-temperature performance. The results indicate that the addition of carbon black improves the asphalt’s stiffness, high-temperature rutting resistance, and low-temperature crack resistance. Moreover, asphalt is more elastic, and its temperature sensitivity is reduced. Raquel et al. [16] found that the mechanical properties of polymer-modified asphalt mixtures with and without carbon black are similar. However, according to the water sensitivity test, the cohesion of the mixture with carbon black is increased. Through semicircular bending test, indirect tensile test, and unconfined compression test, Hamid et al. [9] found that the mechanical strength of carbon black modified asphalt mixture was significantly enhanced compared with that of base asphalt mixture. Li et al. [17] studied the effect of carbon black modifiers on the dynamic modulus of the asphalt mixture. The results revealed that carbon black could effectively improve the anti-deformation ability of the asphalt mixture and reduce its temperature sensitivity. In summary, the above studies show that the addition of carbon black improves the performance of asphalt and asphalt mixtures, especially in terms of high-temperature performance.

Meanwhile, as an abandoned material of industrial production and daily life production, bio-oil has been widely used in pavement engineering research in recent years. Studies [18,19] have shown that mixing bio-oil in asphalt can significantly enhance the low-temperature performance of asphalt and asphalt mixtures. Zeng et al. [20] and Ingrassia et al. [21] found that the asphalt’s corresponding penetration was increased due to the increase in bio-oil content, which indicates the asphalt is softer. Simultaneously, the softening point and ductility of modified asphalt decrease [22] and increase [23], respectively, with the increase in bio-oil content, indicating that the addition of bio-oil can make the asphalt more sensitive to temperature. However, many studies [24,25] have pointed out that bio-oil will weaken the asphalt’s high-temperature performance, which will make the application of bio-oil in pavement engineering more difficult. With the development of technology, composite modification has become a popular technology. Therefore, it is possible to incorporate bio-oil and other materials into the asphalt, not only to enhance the asphalt mixture’s low-temperature crack resistance but also to enhance the asphalt mixture’s high-temperature stability effectively. Lv et al. [26,27] prepared bio-oil/rock asphalt composite modified asphalt. Through dynamic shear rheological and BBR tests, the high-temperature and low-temperature performances of the composite modified asphalt were improved compared with petroleum asphalt. By mixing a certain content of bio-oil into rubber modified asphalt, Lei et al. [28] concluded that modified asphalt’s viscosity first increased and then decreased, and its high-temperature performance was improved. Based on this, the preparation of bio-oil and other materials’ composite modified asphalt and asphalt mixture is the direction of future research.

In this study, carbon black and bio-oil are used as modifiers to be added to the base asphalt. With carbon black content, bio-oil content, and shear time as variables, the BBD method was used for experimental design, and 17 groups of carbon black/bio-oil composite modified asphalt were prepared. Its softening point, penetration, high-temperature performance, and low-temperature performance were tested. In addition, based on the RSM, the carbon black content, bio-oil content, and shear time of the optimal performance carbon black/bio-oil composite modified asphalt were obtained. The conventional base asphalt mixtures and the optimal performance modified asphalt mixtures were formed. Rutting and low-temperature splitting tests were carried out to compare their high-temperature stability and low-temperature resistance. This research can provide a reference for the selection of new green road construction materials.

## 2. Materials and Sample Preparation

### 2.1. Materials

#### 2.1.1. Base Asphalt

The A-70 petroleum asphalt was produced Xiamen Huate group Co., Ltd., Xiamen, China, and was selected as the binder. According to the Chinese standard JTG E20-2011 [29], the main performance indicators of the binder were tested, and the results were shown in Table 1.

#### 2.1.2. Carbon Black

Carbon black mainly includes carbon, hydrogen, and oxygen, of which carbon accounts for about 92% to 99%. The core component of carbon black is an aromatic compound, which is formed by dehydrogenation polymerization and molecular rearrangement for aromatic molecules. As an amorphous carbon, carbon black has good wear resistance and slip resistance, and carbon black has high surface activity and strong adsorption. In addition, carbon black has low cost and abundant sources. Therefore, using carbon black as a modified material can effectively improve the performance of asphalt and asphalt mixtures and reduce the cost of asphalt production.

In this study, N375 rubber carbon black was produced by Tianjin Yiborui Chemical Co., Ltd., Tianjin, China, and was selected as the modifier. The carbon black was made by the thermal cracking method with natural gas as raw material. The characteristics of carbon black were tested with the BK 200C specific surface area and micropore diameter analyzer produced by Beijing Jingwei Gaobo Science and Technology Co., Ltd., Beijing, China. Its particle size was 300 nm, the surface area was 6–10 m^2^/g, and the sulfur absorption value was 11.2 g/kg. The overall degree of aggregation was low, which was beneficial for uniform dispersion in the asphalt.

#### 2.1.3. Bio-Oil

Castor oil was produced by Shenyang Precede Fine Chemical Co., Ltd., Shenyang, China, and was selected as the modifier in this study, which is liquid at homeothermy (25 °C) and has a slightly yellow color. Its main performance indicators were shown in Table 2.

#### 2.1.4. Aggregate

The aggregate was basalt produced in Poly Changda Engineering Co., Ltd., Datong, China. According to the Chinese standard JTG E42-2005 [30], the mechanical indexes of aggregates were tested. The test results were shown in Table 3.

### 2.2. Sample Preparation of Modified Asphalt

Previous studies [14,22] have shown that the optimal content ranges of carbon black and bio-oil are 10–20% and 5–15%, respectively. Within this content range, the rheological and low-temperature properties of modified asphalt were effectively improved. In addition, in this study, the shearing time range is 45–75 min. The Box–Behnken design (BBD) of the response surface method (RSM) was used to design the experiment, which had 17 groups. With carbon black content, bio-oil content, and shear time as the independent variables, the specific design was shown in Table 4.

A high-speed shearing instrument was produced by Anton Paar, Shanghai, China, and was used to prepare carbon black/bio-oil composite modified asphalt with a rotation speed of 5000 r/min through the experimental design. The specific practical steps are as follows: Firstly, place a beaker containing 500 g of base asphalt on the electric furnace (produced by Shanghai Jvjing Precision Instrument Manufacturing Co., Ltd., Shanghai, China.) for 2 h, and control the temperature at 135 °C to make the asphalt fully fluid. Secondly, slowly add the set quality’s carbon black within 10 min, and stir with a glass rod while adding. After the addition of carbon black was completed, use a high-speed shearing instrument to cut for 20 min. Again, slowly add the set quality’s bio-oil within 10 min. Finally, after the modifier’s addition was completed, the sample was sheared with a high-speed shearing instrument for the pre-set time (45 min, 60 min, 75 min).

### 2.3. Sample Preparation of Modified Asphalt Mixture

Based on the Chinese standard JTG F40-2004 [31], the dense grading AC-13 was employed in this paper, and its design grading curve was shown in Figure 1.

The indoor Marshall test method was used to determine the base asphalt mixture’s optimum asphalt content and the optimal performance composite modified asphalt mixture. By testing the Marshall specimen’s volume parameters, the optimal asphalt contents were calculated to be 5.13% and 5.26%, respectively. After that, Marshall samples and rut plates were prepared for the asphalt mixture’s road performance test.

## 3. Methodology and Testing

### 3.1. Response Surface Method (RSM)

The RSM [32] solves the nonlinear multivariate data relationship between the response value and the influencing factors by establishing the relationship between the independent and dependent variables. Therefore, the multivariate nonlinear relationship between the dependent and independent variables can be found. Then, the response surface graph was obtained by simulating the equation, and the combination of the optimal influencing factors was obtained by comparing and analyzing the equation comprehensively.

The least square method was the basic principle of the RSM. It was assumed in advance that the response value *z* and the influencing factor variables $x_{1},x_{2}\cdot \cdot \cdot ,x_{k}$ conform to the following relationship: (1)$$z=f\left(x_{1},x_{2}\cdot \cdot \cdot ,x_{k}\right)+\varepsilon$$
where *ε* is the random variable.

If the response value and the influencing factors meet the approximate linear relationship, the above formula was expanded with the first-order Taylor equation: (2)$$z=\beta_{0}+\beta_{1}x_{1}+\beta_{1}x_{2}+\cdot \cdot \cdot +\beta_{k}x_{k}+\varepsilon$$
where *β_i_*
_(*i* = 1, 2, ···, *k*)_ is the linear slope of the influencing factor *x_i_*
_(*i* = 1, 2, ···, *k*)_.

If other relationships exist between the response value and the influencing factors, the equation needs to be expanded by second-order Taylor, and the second-order model was fitted: (3)$$z=\beta_{0}+\sum_{i=1}^{k}\beta_{i}x_{i}+\sum_{i=1}^{k}\beta_{ii}x_{i}^{2}+\sum_{i<j}^{k}\beta_{ij}x_{i}x_{j}+\varepsilon$$
where *β_ij_* is the interaction between the independent variables, *β_ii_* is the quadratic effect of the *x_i_*.

It can be expressed in matrix form as:(4)$$z=\beta_{0}+X^{\prime }b+X^{\prime }BX+\varepsilon$$
where **X** and **b** are the regression coefficient matrix, **B** is the *k*-order symmetric matrices.
(5)$$X={\left(x_{1},x_{2},\cdot \cdot \cdot ,x_{k}\right)}^{T},\;b={\left(\beta_{1},\beta_{2},\cdot \cdot \cdot ,\beta_{k}\right)}^{T}$$
(6)$$B=\left[\begin{array}{cccc} \beta_{11}\; & \beta_{12}/2 & \cdot \cdot \cdot \; & \beta_{1k}/2\\ \beta_{21}/2 & \beta_{22} & \cdot \cdot \cdot \; & \beta_{2k}/2\\ \cdot \cdot \cdot & \cdot \cdot \cdot & \cdot \cdot \cdot & \cdot \cdot \cdot \\ \beta_{k1}/2 & \beta_{k2}/2 & \cdot \cdot \cdot & \beta_{kk}\; \end{array}\right]$$

If the response value reached the optimal point, there was a relationship:(7)$$\frac{\partial z}{\partial X}=b+2BX=0$$

The solution **X_0_** of the above formula was called the stable point, then:(8)$$X_{0}=-\frac{1}{2}B^{-1}b$$

Incorporating Equation (8) into Equation (4), the predicted response value at the stable point can be obtained:(9)$$z_{0}=\beta_{0}+\frac{1}{2}{X^{\prime }}_{0}b$$

### 3.2. Performance Testing of Modified Asphalt

#### 3.2.1. Physical Properties Test

The sample’s softening point and penetration were tested in this study according to ASTM D113 and ASTM D5, respectively. The test devices used are SYD-2806H Softening Point Tester and CXS-2801 Penetration Tester produced by Shanghai Qigao Instrument Co., Ltd., Shanghai, China. Because the ring and ball method was intuitive and straightforward, it was used to measure the sample’s softening point. The higher the asphalt softening point, the better the high-temperature performance. Penetration was used to describe the sensitivity of the asphalt to temperature. The higher the penetration, the stronger the temperature sensitivity of the asphalt. The penetration test temperature was 25 °C, the penetration time was 5 s, and the load weight was 100 g. All tests were carried out in three parallel tests.

#### 3.2.2. Multiple Stress Creep and Recovery (MSCR) Test

Previous studies [33] have shown that the rutting factor(*G*^*^/sin*δ*) cannot sufficiently characterize the asphalt’s rutting performance, especially the modified asphalt sensitivity to stress levels. Besides, there was a negative correlation between asphalt rutting performance and *G*^*^/sin*δ*. Based on this, the MSCR test was introduced by the Federal Highway Administration to evaluate the rutting performance of asphalt. Afterward, D’Angelo et al. [34] expanded and improved the test method, and the effectiveness of the test has been thoroughly verified.

According to the AASHTO T350 and ASTM D7405 test standards, the MSCR test was carried out by the MCR302 dynamic shear rheometer produced by Anton Paar, Shanghai, China. Before the MSCR test, all asphalt specimens were aged in a rolling thin film oven (RTFO, produced by Wuxi Marite Technology Co., Ltd., Jiangsu, China) at 64 °C. In addition, two stress levels of 0.1 kPa and 3.2 kPa were selected. There were two main parameters obtained in the MSCR test, including irrecoverable creep compliance (*J_nr_*) and strain recovery rate (*R*).

#### 3.2.3. Bending Beam Rheometer (BBR) Test

Based on ASTM D6648, the BBR test of the Superpave asphalt evaluation system was adopted. The test device adopts the SYD-0627 curved beam rheometer produced by Changzhou Dedu Precision Instrument Co., Ltd., Jiangsu, China. The asphalt’s low-temperature performance was evaluated by the measured stiffness modulus (*S*) and creep rate (*M*) values. The beam specimens were used in the test with dimensions of 125 mm × 6.25 mm × 12.5 mm in length, width, and height, respectively. All specimens were tested after RTFO and pressurized aging vessel (PAV) aging, and the test temperature was −18 °C. The test results were recorded when the loading time was 60 s.

### 3.3. Road Performance Testing of Modified Asphalt Mixture

#### 3.3.1. Rutting Test

The rutting resistance of the modified asphalt mixture at high temperature was measured according to the AASHTO T324-04 and Chinese standard T0719-2011 [29] rutting test method. The test device adopts the SYD-0719C-2 automatic rutting tester produced by Shanghai Changji Geological Instrument Co., Ltd., Shanghai, China. According to the deformation curve automatically recorded by the rutting test, the rutting deformation *d*_1_ and *d*_2_ at 45 min (*t*_1_) and 60 min (*t*_2_) were recorded. Then substitute them to Equation (10) to calculate the dynamic stability (*DS*).
(10)$$DS=\frac{(t_{2}-t_{1})\times N}{d_{2}-d_{1}}\times C_{1}\times C_{2}$$
where *DS* is the asphalt mixture’s dynamic stability, times/mm; *d*_1_ and *d*_2_ are the deformation corresponding to time *t*_1_ and *t*_2_ respectively, mm; *N* is the test wheel’s rolling speed, 42 times/min; *C*_1_ is the correction coefficient, 1.0; *C*_2_ is the specimen’s coefficient, 1.0.

#### 3.3.2. Low-Temperature Splitting Test

The low-temperature splitting test was suitable for studying the low-temperature anti-cracking performance of asphalt mixtures. The loading rate was 1 mm/min, the test temperature was −10 °C ± 0.5 °C, and the strip width was 12.7 mm. The test device adopts the MTS-Landmark, Eden Prairie, MN, USA. Different Marshall splitting test specimens were prepared according to the specification [29]. Before the test, the test specimens were put in a temperature-controlled box at −10 °C for more than 4 h to make them reach the controlled temperature, and then quickly take the test specimens out and put them in the splitting fixture. After that, start the test after the test temperature has stabilized for 15 min, record the maximum load *P_T_* and maximum deformation (*X_T_* or *Y_T_*) at the peak, and the test data shall be processed according to the following equations.
(11)$$R_{T}=0.006287P_{T}/h$$
(12)μ=(0.1350A−1.7940)/(−0.5ψ−0.0314)
(13)$$S_{T}=P_{T}\times (0.27+0.1\mu )/(h\times X_{T})$$
where *R_T_* is the splitting tensile strength, MPa; *μ* is Poisson’s ratio of the material; *Ψ* is the ratio of the vertical deformation to the horizontal deformation of the specimen; *S_T_* is the failure stiffness modulus, MPa; *X_T_* and *Y_T_* are the horizontal and vertical deformations of the specimen corresponding to the maximum failure load respectively, mm.

## 4. Results and Discussion

### 4.1. Performance Test Results of Modified Asphalt

#### 4.1.1. Softening Point and Penetration

Table 5 shows the softening point and penetration test results of carbon black/bio-oil composite modified asphalt. The results were the average values after three parallel tests, as shown in Table 5.

Figure 2 shows the three-dimensional (3D) response surface of the softening point for carbon black/bio-oil composite modified asphalt as a function of influence factors. Figure 2a shows that when the shearing time was fixed, the softening point first increases and then decreases with the increase in the carbon black content, while the changing trend with the increase in the bio-oil content is the opposite. Figure 2b,c shows that when the shear time is 45 min and 75 min, the softening point is higher, but overall, the effect of shear time on the softening point is not significant. Then, it can be obtained that the incorporation of carbon black improves the modified asphalt’s high-temperature performance. By contrast, the incorporation of bio-oil has a negative influence on the modified asphalt’s high-temperature performance. Therefore, choosing an appropriate content of modifier can optimize the high-temperature performance of the composite modified asphalt.

Figure 3 shows the 3D response surface of the penetration of carbon black/bio-oil composite modified asphalt with influence factors. Figure 3a shows that the modified asphalt’s penetration was obviously affected by the bio-oil content when the shearing time is fixed. It increases sharply with the increase in the bio-oil content. However, the penetration was less affected by the carbon black content and shear time, for example, when the shear time and carbon black content are 45 min and 15%, respectively. The penetration is 47.57 dmm when the bio-oil content is 5%, while the penetration is 51.73 dmm and 58.21 dmm when the bio-oil content is 10% and 15%. Compared with 5% bio-oil content, the penetration increased by approximately 8.7% and 22.4%, respectively. This phenomenon shows that incorporating bio-oil enhances the composite modified asphalt’s temperature sensitivity and deteriorates the temperature sensitivity. However, the temperature sensitivity can still be controlled by adjusting the carbon black content and shear time.

#### 4.1.2. J_nr_ and R Values

Table 6 shows the MSCR test results, which are the *J_nr_* and *R* values at the stress levels of 0.1 kPa and 3.2 kPa, respectively.

Figure 4 shows the 3D response surface of carbon black/bio-oil composite modified asphalt *J_nr_* value with influencing factors under two stress levels of 0.1 kPa and 3.2 kPa. Among them, the lower the *J_nr_* value, the stronger the permanent deformation resistance of the binder, that is, the better the rutting resistance.

Figure 4 shows that the *J_nr_* values under the two stress levels have similar changes with the content of carbon black, the content of bio-oil, and the shear time. Figure 4a,d shows that when the shear time is constant, the *J_nr_* value decreases with the increase in carbon black content and increases with the increase in bio-oil content. When the shear time and bio-oil content are 60 min and 5%, and the carbon black content is 10%, the *J_nr_* values under the stress levels of 0.1 kPa and 3.2 kPa are 3.76 kPa^−1^ and 4.74 kPa^−1^, respectively. However, when the black content is 20%, the *J_nr_* values at 0.1 kPa and 3.2 kPa stress levels are 3.31 kPa^−1^ and 4.28 kPa^−1^, respectively, which are reduced by 11.9% and 9.7%, respectively. When the shear time and the carbon black content are 60 min and 15%, and the bio-oil content is 5%, the *J_nr_* values at 0.1 kPa and 3.2 kPa stress levels are 3.47 kPa^−1^ and 4.42 kPa^−1^, respectively. When the amount of bio-oil is 15%, the *J_nr_* values at 0.1 kPa and 3.2 kPa stress levels are 3.58 kPa^−1^ and 4.58 kPa^−1^, respectively, which are 3.2% and 3.6% higher than that. This result indicates that the incorporation of carbon black improves the asphalt’s rutting performance. By contrast, the incorporation of bio-oil has a negative effect on the asphalt’s rutting performance. However, the degree of influence is lower than the positive effect of carbon black. In addition, it can be seen from Figure 4 that, under any stress level, the *J_nr_* value decreases first and then increases with the increase in shear time, indicating that there is an optimal shear time to make the carbon black/bio-oil composite modified asphalt have the optimal high-temperature performance.

Figure 5 shows the 3D response surface of carbon black/bio-oil composite modified asphalt *R* value with influencing factors under two stress levels of 0.1 kPa and 3.2 kPa. The *R* value was used to evaluate the asphalt’s elastic performance. The higher the *R* value, the better the binder’s strain recovery and the elastic performance.

It can be seen from Figure 5 that the *R* value under the two stress levels has similar changes with the content of carbon black, the content of bio-oil, and the shear time. Figure 5a,d shows that, when the shear time is constant, the *R* value increases with the increase in the carbon black content and decreases with the increase in the bio-oil content, and the range of changes with the bio-oil is less than the range of change with carbon black. When the shearing time and bio-oil content are fixed, and the carbon black content is increased from 10% to 20%, the *R* value under the stress level of 0.1 kPa and 3.2 kPa increases by 5–6%. When the shearing time and the content of carbon black are fixed, when the content of bio-oil is increased from 5% to 15%, the *R* value under the stress level of 0.1 kPa and 3.2 kPa is reduced by about 1.5–2.5%. This result indicates that incorporating carbon black improves the elastic strain recovery rate of the asphalt binder. By contrast, bio-oil incorporation reduces the elastic strain recovery rate of the asphalt binder, but its effect is insignificant. In addition, it can be seen from Figure 5 that, no matter what the stress level, the longer the shearing time, the greater the *R* value. The reason is that the longer the shearing time is, the better the carbon black and bio-oil can be dispersed in the asphalt binder, and the higher the uniformity will be.

In summary, from the test results of *J_nr_* and *R* values, although the incorporation of bio-oil has a negative influence on the asphalt’s high-temperature performance, the incorporation of carbon black can effectively compensate for this defect. From the test results in Table 6, it can be seen that whether the stress level is 0.1 kPa or 3.2 kPa, the *J_nr_* value of the carbon black/bio-oil composite modified asphalt is lower than that of the base asphalt, and the *R* value is greater than that of the base asphalt. Generally, the high-temperature performance of the carbon black/bio-oil composite modified asphalt is improved compared with the base asphalt’s high-temperature performance. Therefore, using carbon black/bio-oil composite modified asphalt as a binder in actual pavement engineering can be considered.

#### 4.1.3. S and M Values

Table 7 shows the BBR test results, which are the measured stiffness modulus *S* and creep rate *M* values.

Figure 6 shows the 3D response surface of the *S* and *M* values of carbon black/bio-oil composite modified asphalt with test influencing factors. It can be seen from Figure 6 that all tests meet the specification requirements (*S*_60s_ ≤ 300 MPa, *M*_60s_ ≥ 0.300) [29]. Among them, the creep stiffness modulus *S* is used to evaluate the asphalt’s stiffness, and the smaller the *S* value, indicating that the smaller the stress generated by the unit strain, the softer the asphalt. The creep recovery rate *M* value is used to evaluate the stress relaxation ability of asphalt. The larger the *M* value, the better the stress relaxation ability and its crack resistance.

It can be obtained from Figure 6 that when the shear time and the carbon black content are constant, with the increase in the bio-oil content, the *S* value gradually decreases. By contrast, the *M* value increases significantly, and the increase rate gradually slows down. Taking the shear time of 60 min as an example, when the carbon black content is 10% and the content of bio-oil is increased from 5% to 15%, the *S* value is reduced by about 170 MPa, and the *M* value is increased by about 0.084. When the carbon black content is 20% and the bio-oil content is increased from 5% to 15%, the *S* value is reduced by about 90 MPa, and the *M* value is increased by about 0.061. Table 7 also shows that, except when the bio-oil content is 5%, the *S* value of the carbon black/bio-oil composite modified asphalt is lower than the *S* value of the base asphalt, and the *M* value is higher than the *M* value of the base asphalt. The results show that the incorporation of bio-oil significantly improves the asphalt’s low-temperature performance. However, the incorporation of carbon black will impact the ability of bio-oil to improve the asphalt’s low-temperature performance, and the interaction between them may be causing this phenomenon. Besides, Figure 6 shows that when the content of bio-oil or carbon black is fixed, with the increase in shear time, the *S* value first decreases and then increases, while the *M* value first increases and then decreases. This phenomenon indicates an optimal shear time to optimize the carbon black/bio-oil composite modified asphalt’s low-temperature performance.

### 4.2. Performance Optimization Based on RSM Model

Based on the RSM, a quadratic function model was used to fit the test influencing factors and modified asphalt performance indicators, and the fitting models were as shown in Equations (14)–(21). Table 8 shows the variance analysis results of the fitted model.
(14)$$\begin{array}{l} Softing\;point=51.411+0.872A+0.198B-0.333C+0.020AB\\ +0.0009AC+0.0014BC-0.0375A^{2}-0.0355B^{2}+0.00247C^{2} \end{array}$$
(15)$$\begin{array}{l} Penetration=57.6-0.973A+0.225B-0.12C-0.017AB\\ +0.0036AC-0.0018BC+0.026A^{2}+0.062B^{2}+0.0003C^{2} \end{array}$$
(16)$$\begin{array}{l} J_{nr0.1}=6.711-0.202A+0.134B-0.06C+0.0002AB\\ +0.0008AC-0.0007BC+0.0035A^{2}-0.004B^{2}+0.0004C^{2} \end{array}$$
(17)$$\begin{array}{l} J_{nr3.2}=7.521-0.196A+0.138B-0.057C+0.0001AB\\ +0.0008AC-0.0006BC+0.0034A^{2}-0.0044B^{2}+0.0004C^{2} \end{array}$$
(18)$$\begin{array}{l} R_{0.1}=22.77-0.636A+0.065B+0.132C+0.01AB\\ -0.00007AC-0.008BC+0.0365A^{2}+0.0035B^{2}-0.0002C^{2} \end{array}$$
(19)$$\begin{array}{l} R_{3.2}=18.098-0.766A+0.051B+0.17C+0.009AB\\ -0.0003AC-0.008BC+0.0417A^{2}+0.0037B^{2}-0.0005C^{2} \end{array}$$
(20)$$\begin{array}{l} S=1363-27.925A-45.1B-24.275C+0.79AB\\ -0.187AC+0.203BC+0.95A^{2}+0.38B^{2}+0.206C^{2} \end{array}$$
(21)$$\begin{array}{l} M=0.193-0.008A+0.035B+0.0036C-0.0002AB\\ +0.00008AC-0.00006BC+0.0002A^{2}-0.001B^{2}-0.0004C^{2} \end{array}$$
where *A* is the carbon black content, %; *B* is the bio-oil content, %; *C* is the shear time, min.

Table 8 shows that the Adjusted R^2^ of all models is above 0.96, indicating that this model can effectively and reliably fit the test results. Therefore, by customizing the dependent variable’s target value, the value of the independent variable and other indicators can be obtained. Based on the analysis of the above test results, the carbon black/bio-oil composite modified asphalt has good high-temperature and low-temperature performances. It can be used in paving roads in different climate regions. Therefore, the corresponding high-temperature or low-temperature indexes can be selected as the target value according to the hot or cold climate, and other indexes meet the design requirements. Meanwhile, in this work, the optimal contents of carbon black and bio-oil are 15.05% and 9.631%, respectively; the shear time is 62.667 min; the corresponding softening point and penetration are 48.043 °C and 48.043 dmm; and *J_nr_*_0.1_ and *J_nr_*_3.2_ are 3.598 kPa^−1^ and 4.58 kPa^−1^. In addition, *R*_0.1_ and *R*_3.2_ are 26.589% and 22.031%, and the *S* value and *M* value are 106.096 MPa and 0.441, respectively.

### 4.3. Road Performance Test Results of Modified Asphalt Mixture

#### 4.3.1. High-Temperature Stability

Table 9 and Figure 7 show the *DS* results of the base asphalt mixture and the carbon black/bio-oil composite modified asphalt mixture. Five parallel tests were carried out for each group of tests.

Figure 7 shows the *DS* of the two types of asphalt mixture that meet the specification requirements. The high-temperature stability of the carbon black/bio-oil composite modified asphalt mixture is better than that of the base asphalt mixture, and its *DS* is about twice that of the base asphalt mixture. This phenomenon shows that the appropriate content of carbon black and bio-oil can effectively improve the high-temperature stability of the asphalt mixture.

#### 4.3.2. Low-Temperature Crack Resistance

Table 10 and Figure 8 show the results and coefficient of variation (*C*_v_) of the low-temperature splitting test of base asphalt mixture and carbon black/bio-oil composite modified asphalt mixture at −10 °C, including splitting tensile strength, failure stiffness modulus, and failure tensile strain. For each group of experiments, five parallel experiments were performed.

It can be seen from Figure 8 that the splitting tensile strength, failure stiffness modulus, and failure tensile strain of the carbon black/bio-oil composite modified asphalt mixture were all greater than those of the base asphalt mixture, and these indexes were increased by 18.9%, 5.2%, and 26.4%, respectively. The results show that the incorporation of carbon black and bio-oil improves the toughness and deformation resistance of the asphalt mixture at low-temperature, thereby improving the asphalt mixture’s low-temperature crack resistance.

### 4.4. Economic Analysis

Based on the optimized results determined by RSM, 0.1505 kg of carbon black and 0.09631 kg of bio-oil are needed to prepare 1 kg of carbon black/bio-oil composite modified asphalt. Currently, the price of carbon black, bio-oil, and A-70 petroleum asphalt on the market is about 0.26 USD/kg, 0.30 USD/kg, and 0.55 USD/kg. Hence, the production of 1 kg of modified asphalt needs to consume about 0.48 USD in raw materials, while the price of A-70 petroleum asphalt is 0.55 USD, and the price of SBS modified asphalt that is often used at present is about 0.57 USD. Compared with the price of raw materials, the raw material price of 1 kg carbon black/bio-oil modified asphalt is about 14.5% and 18.8% lower than that of base asphalt and SBS modified asphalt, respectively. From the perspective of sustainable development, on the one hand, carbon black/bio-oil modified asphalt has good economic benefits, and its production cost is low. On the other hand, the raw materials of carbon black/bio-oil modified asphalt are all waste materials. It has played a crucial role in environmental protection. Therefore, this modified asphalt is worthy of further study.

## 5. Conclusions

In this study, carbon black/bio-oil composite modified asphalt was prepared by adding carbon black and bio-oil to the base asphalt. Through softening point, penetration, MSCR, and BBR tests, the modified asphalt’s physical properties, high-temperature performance, and low-temperature performance were tested. The modifier’s optimal content and the optimal shear time were determined based on the RSM. In addition, the base asphalt mixture and the carbon black/bio-oil composite modified asphalt mixture with optimal performance were formed, and the rutting test and the low-temperature splitting test were carried out. Then, the high-temperature stability and low-temperature crack resistance of the two mixtures were compared and analyzed. The main conclusions were as follows:

(1) The incorporation of carbon black and bio-oil has a significant influence on the asphalt’s high-temperature and low-temperature performance. Among them, carbon black can effectively enhance the asphalt’s high-temperature performance, while bio-oil can enhance the asphalt’s low-temperature performance.

(2) Based on the response surface method, the performance indicators of carbon black/bio-oil composite modified asphalt show a quadratic function model with carbon black, bio-oil content, and shear time. In addition, the fitting correlation coefficients are all above 0.96. For the optimal performance of carbon black/bio-oil composite modified asphalt, the carbon black and bio-oil contents are 15.05% and 9.631%, respectively, and the shear time is 62.667 min.

(3) The high-temperature stability and low-temperature crack resistance of the carbon black/bio-oil composite modified asphalt mixture are significantly better than those of the base asphalt mixture, which can provide a reference for the selection of new road construction environmentally friendly materials.

(4) In this work, although some performances of carbon black/bio-oil composite modified asphalt and its mixture were studied, they were all macroscopic properties. In future work, the analysis should be combined with microscopic performance tests. Fourier transform infrared spectroscopy, four-component analysis, CT, and scanning electron microscopy tests should be performed to explore the properties of carbon black/bio-oil composite modified asphalt and its mixture in more depth.

## Figures and Tables

**Figure 1 materials-14-04910-f001:**
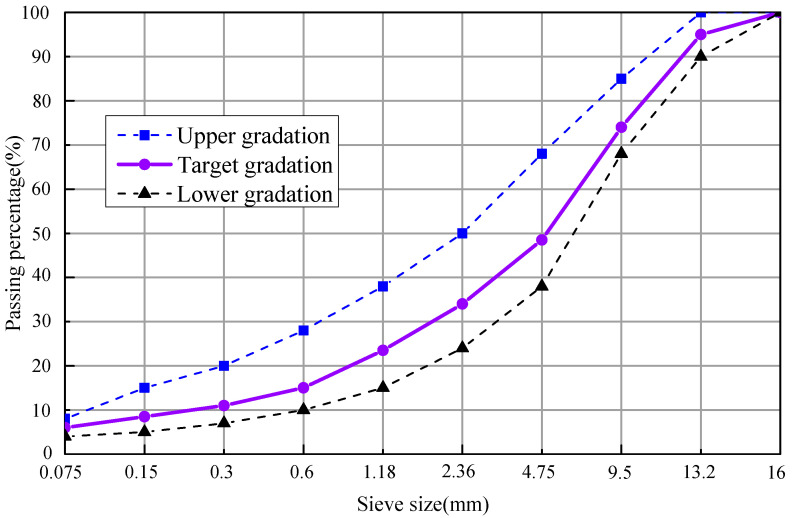
The design grading curve.

**Figure 2 materials-14-04910-f002:**
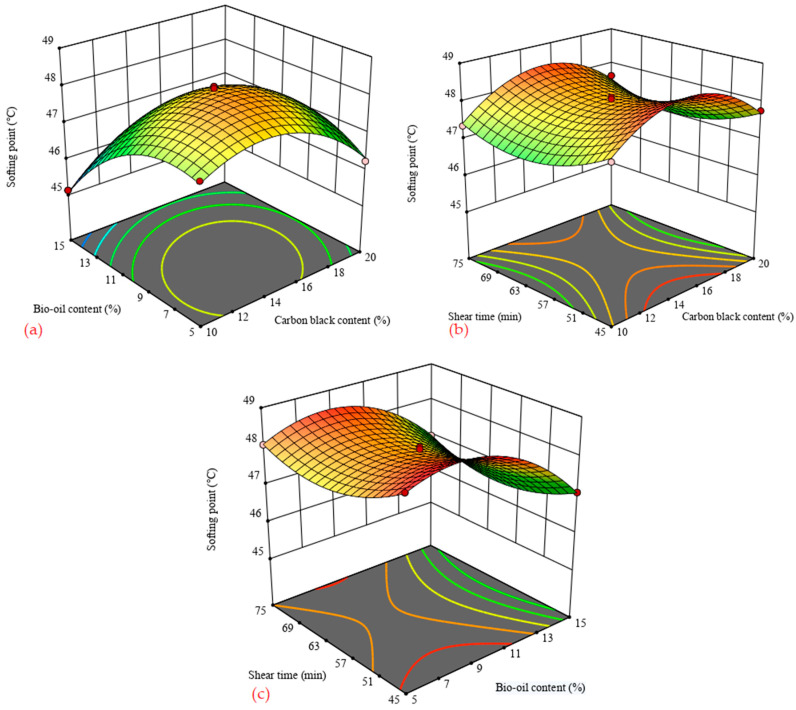
3D surface of softening point with influence factors. (**a**) shear time = 60 min; (**b**) bio-oil content = 10%; (**c**) carbon black content = 15%.

**Figure 3 materials-14-04910-f003:**
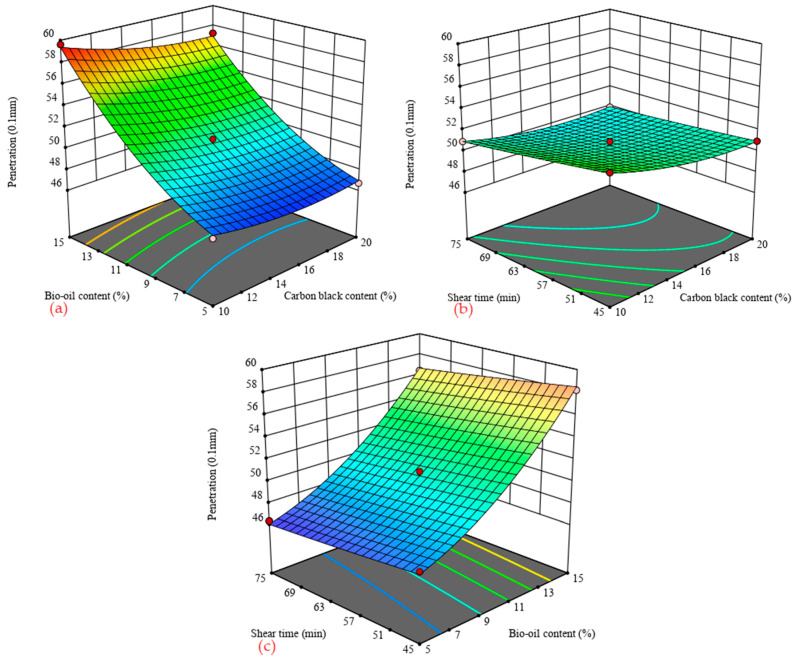
3D surface of penetration with influence factors. (**a**) shear time = 60 min; (**b**) bio-oil content = 10%; (**c**) carbon black content = 15%.

**Figure 4 materials-14-04910-f004:**
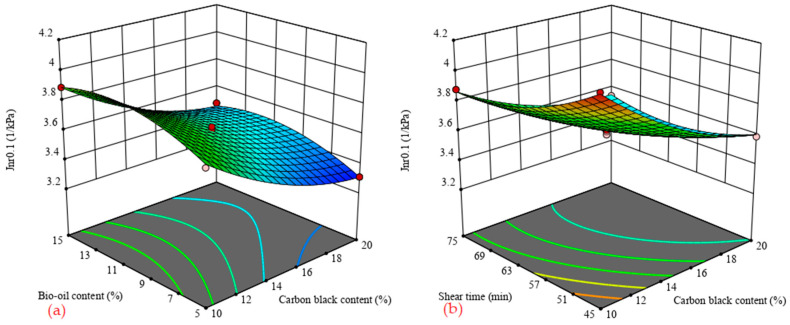
3D surface of *J_nr_* value with influence factors. (**a**) shear time = 60 min; (**b**) bio-oil content = 10%; (**c**) carbon black content = 15%; (**d**) shear time = 60 min; (**e**) bio-oil content = 10%; (**f**) carbon black content = 15%.

**Figure 5 materials-14-04910-f005:**
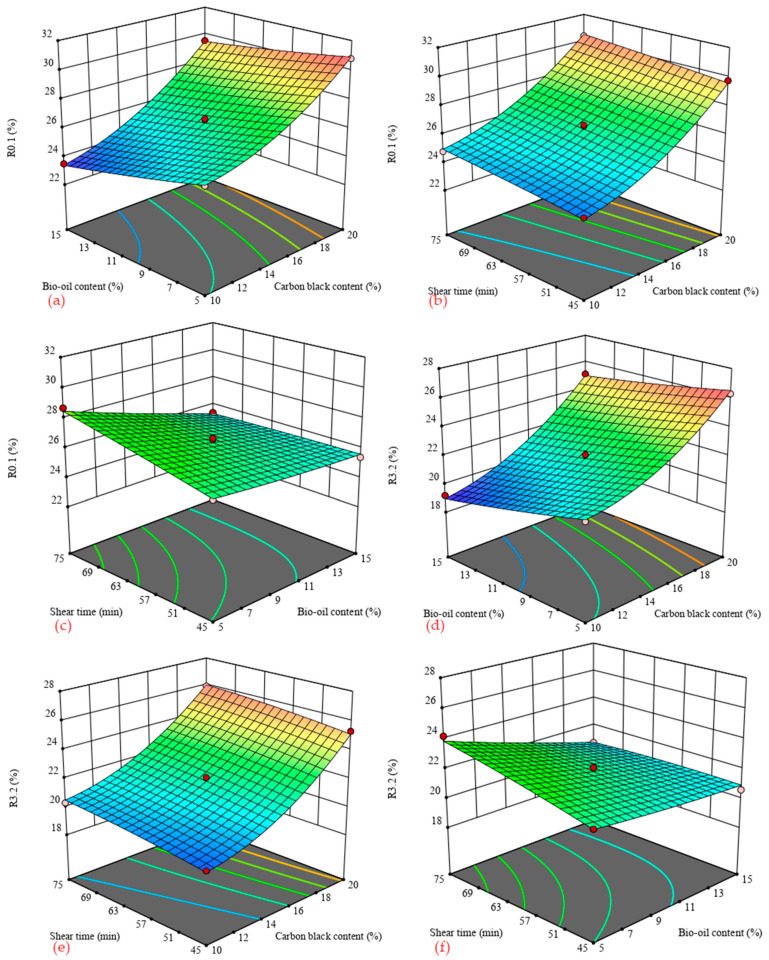
3D surface of *R* value with influence factors. (**a**) shear time = 60 min; (**b**) bio-oil content = 10%; (**c**) carbon black content = 15%; (**d**) shear time = 60 min; (**e**) bio-oil content = 10%; (**f**) carbon black content = 15%.

**Figure 6 materials-14-04910-f006:**
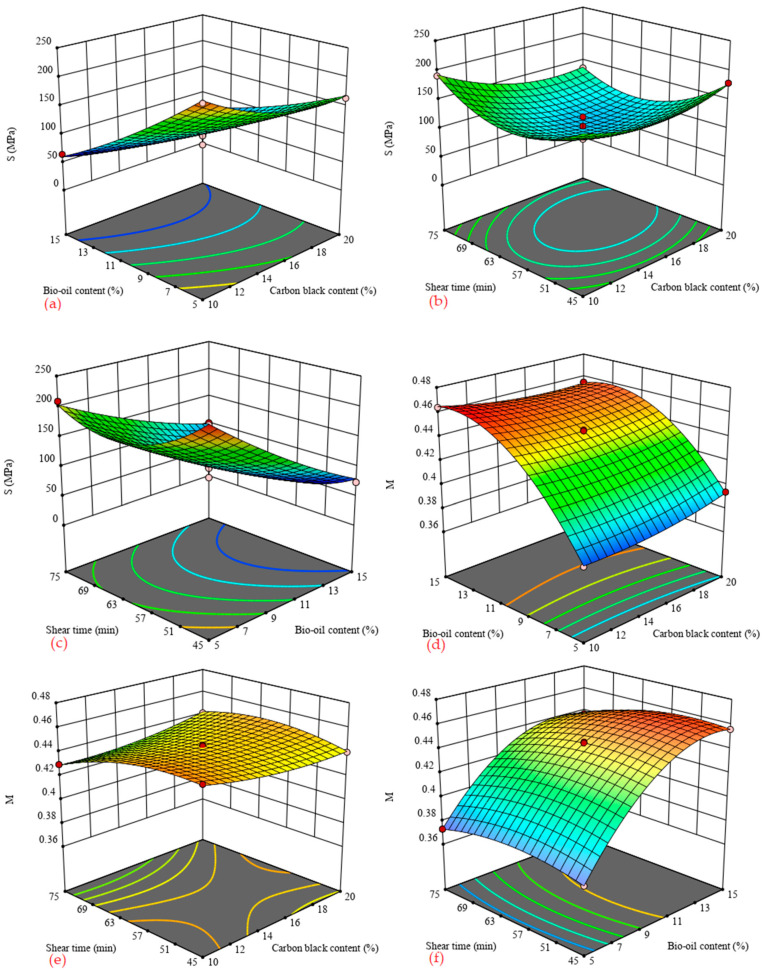
3D surface of *S* and *M* values with influence factors. (**a**) shear time = 60 min; (**b**) bio-oil content = 10%; (**c**) carbon black content = 15%; (**d**) shear time = 60 min; (**e**) bio-oil content = 10%; (**f**) carbon black content = 15%.

**Figure 7 materials-14-04910-f007:**
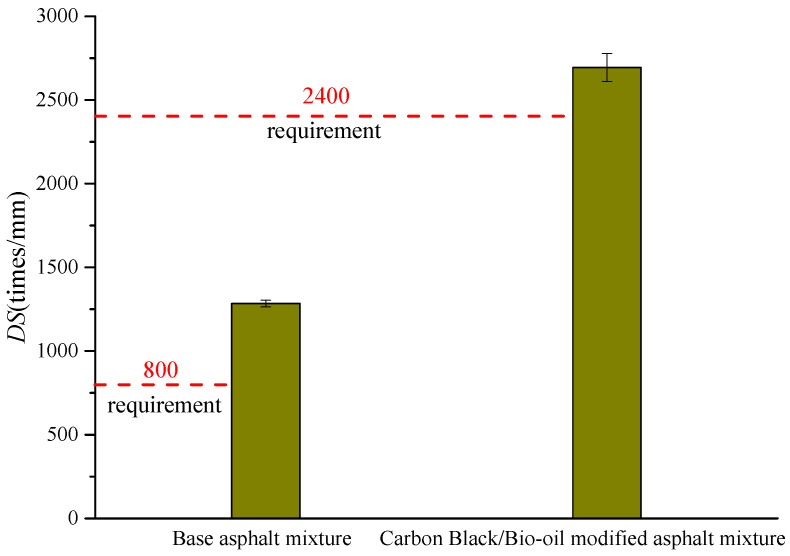
Comparison of *DS* results of different asphalt mixtures.

**Figure 8 materials-14-04910-f008:**
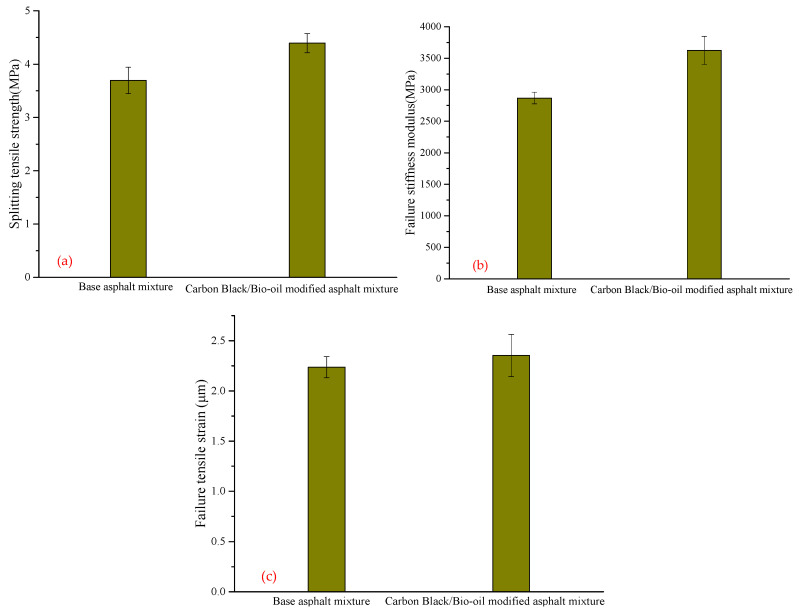
Comparison of low-temperature splitting test results of different asphalt mixtures. (**a**) Splitting tensile strength; (**b**) Failure stiffness modulus; (**c**) Failure tensile strain.

**Table 1 materials-14-04910-t001:** Performance indicators of A-70 petroleum asphalt.

Performance Indicators	Unit	Test Results	Requirement [29]	Test Methods
Softening point (ring and ball method)	°C	48.5	≥46	T0606-2011
Penetration (25 °C, 100 g, 5 s)	0.1 mm	64	60–80	T0604-2011
Ductility (15 °C, 5 cm/min)	cm	>100	≥100	T0605-2011
Dynamic viscosity (60 °C)	Pa·s	199.3	≥180	T0620-2000
Wax content	%	1.9	≤2.2	T0615-2011
Density (15 °C)	g/cm^3^	1.029	-	T0603-2011

**Table 2 materials-14-04910-t002:** Performance indicators of castor oil.

Performance Indicators	Unit	Test Results
Relative density	g/cm^3^	0.955
Refractive index	-	1.477
Viscosity (20 °C)	Pa·s	15.3
Freezing point	°C	−10
Ignition point	°C	322

**Table 3 materials-14-04910-t003:** The mechanical indexes of basalt.

Mechanical Indexes	Crush Value (%)	Polishing Value (BPN)	Abrasion Value (%)
Test results	18.8	55.2	17.2
Technical requirement [30]	≤30	≥45	≤30
Test method	T0316-2005	T0321-2005	T0317-2005

**Table 4 materials-14-04910-t004:** Experimental design based on BBD method.

Run	Carbon Black Content (%)	Bio-Oil Content (%)	Shear Time (min)
1	20	15	60
2	20	10	75
3	20	10	45
4	10	10	75
5	15	15	75
6	15	10	60
7	15	5	75
8	15	5	45
9	10	10	45
10	10	15	60
11	20	5	60
12	15	10	60
13	15	10	60
14	10	5	60
15	15	10	60
16	15	10	60
17	15	15	45

**Table 5 materials-14-04910-t005:** Average value of the softening point and penetration test results.

Run	Carbon Black Content (%)	Bio-Oil Content (%)	Shear Time (min)	Softening Point (°C)	Penetration (0.1 mm)
1	20	15	60	45.96	57.24
2	20	10	75	47.64	49.89
3	20	10	45	47.76	50.98
4	10	10	75	47.36	50.94
5	15	15	75	46.98	56.42
6	15	10	60	47.89	50.46
7	15	5	75	48.06	46.32
8	15	5	45	48.59	47.57
9	10	10	45	47.75	53.12
10	10	15	60	45.13	59.62
11	20	5	60	46.25	46.69
12	15	10	60	48.07	50.93
13	15	10	60	48.12	50.44
14	10	5	60	47.4	47.32
15	15	10	60	47.99	50.33
16	15	10	60	47.98	50.48
17	15	15	45	47.08	58.21

**Table 6 materials-14-04910-t006:** MSCR test results.

Run	Carbon Black Content (%)	Bio-Oil Content (%)	Shear Time (min)	*J_nr_*_0.1_ (kPa^−1^)	*J_nr_*_3.2_ (kPa^−1^)	*R*_0.1_ (%)	*R*_3.2_ (%)
1	20	15	60	3.48	4.44	29.5	25.1
2	20	10	75	3.54	4.51	30.4	25.9
3	20	10	45	3.59	4.57	29.8	25.3
4	10	10	75	3.88	4.86	24.8	20.3
5	15	15	75	3.52	4.51	25.4	20.7
6	15	10	60	3.61	4.56	26.2	21.5
7	15	5	75	3.51	4.48	28.7	24.2
8	15	5	45	3.59	4.55	26.3	21.8
9	10	10	45	4.18	5.15	24.1	19.6
10	10	15	60	3.89	4.87	23.5	19.2
11	20	5	60	3.31	4.28	30.8	26.3
12	15	10	60	3.62	4.6	26.7	22.1
13	15	10	60	3.59	4.59	26.3	21.8
14	10	5	60	3.74	4.72	25.8	21.3
15	15	10	60	3.63	4.65	26.6	22.1
16	15	10	60	3.64	4.6	26.2	21.7
17	15	15	45	3.81	4.77	25.4	20.6
Base asphalt	0	0	-	4.21	5.22	21.8	18.4

**Table 7 materials-14-04910-t007:** The BBR test results.

Run	Carbon Black Content (%)	Bio-Oil Content (%)	Shear Time (min)	*S* (MPa)	*M*
1	20	15	60	77	0.455
2	20	10	75	136	0.441
3	20	10	45	179	0.440
4	10	10	75	191	0.430
5	15	15	75	97	0.437
6	15	10	60	82	0.445
7	15	5	75	209	0.373
8	15	5	45	247	0.375
9	10	10	45	178	0.453
10	10	15	60	65	0.464
11	20	5	60	164	0.394
12	15	10	60	98	0.445
13	15	10	60	105	0.446
14	10	5	60	231	0.380
15	15	10	60	99	0.444
16	15	10	60	121	0.445
17	15	15	45	74	0.456
Base asphalt	0	0	-	198	0.381

**Table 8 materials-14-04910-t008:** ANOVA for model.

Fit Statistics	*Softening Point*	*Penetration*	*J_nr_* _0.1_	*J_nr_* _3.2_	*R* _0.1_	*R* _3.2_	*M*	*S*
*p*-value	<0.0001	<0.0001	<0.0001	<0.0001	<0.0001	<0.0001	<0.0001	<0.0001
F-value	124.72	259.2	137.41	95.46	140.54	84.19	44.27	1591.9
Lack of fit *p*-value	0.249	0.128	0.276	0.824	0.409	0.233	0.849	0.136
Lack of fit F-value	2.06	3.51	1.87	0.30	1.23	2.18	0.26	3.36
Adjusted R²	0.986	0.993	0.987	0.982	0.987	0.979	0.961	0.999
Predicted R²	0.936	0.964	0.944	0.966	0.953	0.904	0.932	0.994
Adeq Precision	42.99	52.06	49.83	41.34	39.84	30.76	20.63	119.44

**Table 9 materials-14-04910-t009:** *DS* results of different asphalt mixtures.

Asphalt Mixture Type	Test Results (times/mm)	Average Value (times/mm)	Coefficient of Variation (%)	Requirement [29]
Base	1270/1258/1296/1307/1288	1284	1.54	≥800
Carbon black/bio-oil composite modified	2678/2745/2693/2788/2567	2694	3.09	≥2400

**Table 10 materials-14-04910-t010:** Low-temperature splitting test results of different asphalt mixtures.

Asphalt Mixture Type	Splitting Tensile Strength (MPa)/*C*_v_ (%)	Failure Stiffness Modulus (MPa)/*C*_v_ (%)	Failure Tensile Strain (μm)/*C*_v_ (%)
Base	3.696/6.66	2867/3.23	2.236/4.47
Carbon black/bio-oil composite modified	4.394/4.05	3625/6.15	2.353/8.93

## Data Availability

The data is available on request from the corresponding author.

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
