# Peer review of "Laboratory Investigation of Carbon Black/Bio-Oil Composite Modified Asphalt"

_materials, 2021, doi:10.3390/ma14174910_

Round 1

Reviewer 1 Report

Paper title: Laboratory Investigation of Carbon Black/Bio-oil Composite Modified Asphalt (ID: MATERIALS 1339461) Recommendation: Reject The paper deals with the feasibility of using environmental-friendly materials such as carbon black and bio-oil as bitumen modifiers with the aim of improving both binder phase and the corresponding asphalt mixture. In particular, the Authors considered the carbon black and bio-oil contents as well as shear time (used during the production of the modified bitumen) as the main parameters affecting the mechanical response of the modified bitumens and the Response Surface Method (RSM),within the Box-Behnken design, was used as tool to analyse the test results. The topic is relevant and timely, considering that nowadays the search for any base materials and/or technology alternative to the “traditional ones” which are consistent with the pavement sustainability is the primary interest all over the world. Notwithstanding, it is believed that the research study as well as the relevance of results show some limitations that do not enable, on the one hand, to support satisfactorily the approach proposed by the Authors, on the other hand, to strengthen the background on the use of “green materials” to modify the bituminous materials. The main concerns and reservations are summarized as follows: - The Authors selected the carbon black and bio-oil content as well as the shear time as influencing factors. In particular, the shear time is an important variable especially in the case of the production of “composite binder” because it affect the dispersion of the addition within the bituminous matrix and thus the homogeneity of the final product. From a general point of view, this means that depending on the type of “addition”, the system can reach a proper homogeneity fast (requiring lower shear time) or more slowly (requiring higher shear time). Moreover, a temporary homogeneity (obtained during and at the end of the production phase) does not assure a stability condition of the sample. As conclusion, the Authors would have to analyse the effect of the shear time in terms of storage stability of the modified bitumen (Authors never mentioned this important aspect considering that it is necessary condition for the modified bitumen). Moreover, it is believed a bit particular that the shear time effects are random (performance first decreases and then increases, or viceversa) depending on the mechanical parameter considered. - The section “Results and Discussion” about the modified asphalt does not show any sound and real result analysis, but a simple presentation of experimental data is presented supported by general comments and sometimes, obvious conclusions. I mean that, in general, the “stiffening effect” of a solid modifier (carbon black) and the “softening effect” of a liquid modifier (bio-oil) as well as their combination on the response of bitumen has been extensively recognized and investigated. Therefore, the fact that physical properties (penetration, softening point) as well as rheological ones (non-recoverable creep compliance, strain recovery percentage, creep stiffness and creep rate) changed according to the increased or decreased stiffness of the modified bitumen produced with carbon black and bio-oil was strongly expected. Indeed, the Authors never referred to any potential physical-chemical interaction between the modifiers each other and/or with bitumen in order to support the experimental findings. - Despite the response surface methodology is a powerful statistical tool to find the relationships between several explanatory variables and several response variables, it is believed that this approach is “unnecessary”, considering that the rank of the effects due to the single independent variables is already evident from the set of the experimental data. Moreover, it is right that the RSM allowed to find quadratic function models fitting the independent and dependent variables, however the practical utility of these laws is unknown because the validity of the found “optimal solution” can be only limited to the investigated materials and related boundary condition (e.g. shear time). - The Authors stated that “the carbon black/bio-oil composite modified asphalt has good high-temperature and low-temperature performances.”, but it is unusual that no comparison with the performance of the reference bitumen (carbon black and bio-oil content = 0) was showed, highlighting the “potential improvement” due to addition of the modifiers. - The Authors rightfully mention sustainability and advantages that may be attributed to modified materials like this, nevertheless the paper lack in a proper discussion on the economic aspects of proposed modification (things like added cost, availability of the material, mass production requirements, and so on) which would have been beneficial and support the “sustainable feature” of the topic discussed in the paper. - It is believed that in the Section “Road performance test results of modified asphalt mixture“ the mere comparison of the selected two mechanical parameters between the reference and the modified mixtures is not sufficiently exhaustive to “exalt” the enhanced performance of the carbon black/bio-oil modified mixture. I mean that the improved performance of modified mixture was likely expected with respect the reference one, indeed the comparison with a reference “conventional” modified asphalt mixture (e.g. SBS modified mixture) would have allowed or not to properly highlight the potential performance benefits of the investigated “green modified mixture”. - It is important to highlight that, within an international research scenario, as specific standardized test protocol are performed it is right to cite international reference standards (e.g. ASTM, AASHTO, CEN and so on) and not merely to cite “local technical specifications”, this source could be unknown.

Reviewer 2 Report

The manuscript presents the results of  carbon black and castor oil modification road bitumen type 60/80. The materials used: bitumen, carbon black, castor oil, mineral aggregates were characterized.

The methods/devices by which the properties of carbon black were determined - the particle size, the specific surface, etc. are not mentioned. (rows 120… 123). I don't see the connection between castor oil and homeothermy (line 126). The devices used, manufacturer and year were not clearly mentioned.

The authors noted that the introduction of castor oil has a negative effect on performance at high temperatures, diminished by the addition of carbon black.   For a better understanding of the phenomena before and after asphalt modification, a bitumen composition analysis by compound classes (iatroscan S.A.R.A. or the classical method by solvent separation) is very useful. Thus, the results obtained would be clearer .

The conclusions clearly present the results obtained by the authors. The bibliography is recent and well structured. The manuscript presents interesting data, useful from an applicative point of view, reason for which I recommend its publication after remedying the ones presented above.

Round 2

Reviewer 1 Report

It is believed that slight concerns about some parts of the paper still remains, however the paper was improved with useful and pertaining information which made the paper clearer and more sound.

I think that the paper has gained a chance to be published.